# Changes in Oxygenation and Serological Markers in Acute Exacerbation of Interstitial Lung Disease Treated with Polymyxin B Hemoperfusion

**DOI:** 10.3390/jcm11092485

**Published:** 2022-04-28

**Authors:** Song-I Lee, Chaeuk Chung, Dongil Park, Da Hyun Kang, Jeong Eun Lee

**Affiliations:** Division of Pulmonary and Critical Care Medicine, Department of Internal Medicine, Chungnam National University Hospital, Chungnam National University School of Medicine, Munhwa-ro, Jung-gu, Daejeon 35015, Korea; newcomet01@naver.com (S.-I.L.); universe7903@gmail.com (C.C.); rahm3s@gmail.com (D.P.); ibelieveu113@cnuh.co.kr (D.H.K.)

**Keywords:** interstitial lung disease, idiopathic pulmonary fibrosis, polymyxin B, biochemical marker, prognosis

## Abstract

Background: Polymyxin B direct hemoperfusion (PMX-DHP) has been tried in acute exacerbation of interstitial lung disease (AE-ILD) patients and has shown clinical benefit. In this study, we tried to investigate the change in oxygenation and serologic markers after PMX-DHP treatment in AE-ILD patients in Korea. Methods: We reviewed the medical records of twenty-two patients who were admitted for AE-ILD and underwent PMX-DHP treatment. Changes in vital signs and laboratory findings before and after treatment were compared and factors related to 90-day mortality were analyzed using the Cox regression model. Results: Of the 22 included patients, 11 (50%) patients were diagnosed with idiopathic pulmonary fibrosis. In AE-ILD patients treated with PMX-DHP, the 28-day mortality rate was 45.5% and the 90-day mortality rate was 72.7%. The P/F ratio before and after PMX-DHP treatment significantly improved in patients from baseline to 24 h (median (IQR), 116.3 (88.5–134.3) mmHg vs. 168.6 (115.5–226.8) mmHg, *p* = 0.001), and 48 h (116.3 (88.5–134.3) mmHg vs. 181.6 (108.9–232.0) mmHg, *p* = 0.003). Also, white blood cells (WBCs) and C-reactive protein (CRP) were decreased after PMX-DHP treatment. High acute physiology and chronic health evaluation II scores were associated with 90-day mortality. Conclusions: In patients with AE-ILD, PMX-DHP treatment was associated with an improved P/F ratio and lower WBC and CRP levels.

## 1. Introduction

Interstitial lung disease (ILD) is a lung disease associated with high morbidity and mortality with diffuse parenchymal lung disease [1]. Idiopathic pulmonary fibrosis (IPF) is a type of ILD of unknown etiology, characterized by chronic progressive fibrosis, and the median survival time is known to be about 3–5 years after diagnosis [2]. IPF patients have a variable course, and about one-third of patients experience an acute exacerbation with rapid progression of dyspnea within 1 month. Approximately half of patients who experience an acute exacerbation (AE) of IPF have been reported to result in respiratory failure and death during hospitalization. The average survival time from the onset of an acute exacerbation varies, but some studies report the average survival period is 3–13 days, with a high mortality rate up to 85% [3,4,5,6]. In addition, a poor prognosis has been reported in other types of acute or subacute ILD [1,7]. In acute exacerbations of IPF or ILD, salvage therapy other than high-dose steroids and treatment including anti-inflammatory drugs and immunosuppressants showed little effect on the improvement of prognosis [3,8]. Therefore, there has been a continuous demand for the development of a new method that can show a favorable effect in the treatment of AE of IPF.

Polymyxin B direct hemoperfusion (PMX-DHP) is a medical device using polystyrene fibers and it was originally developed to remove endotoxins in endotoxemia observed in sepsis caused by Gram-negative bacilli [9], and in acute respiratory distress syndrome (ARDS). PMX-DHP is also effective in reducing several other serological markers such as interleukin (IL)-6, IL-9, IL-12, IL-17, platelet-derived growth factor, vascular endothelial growth factor and tumor necrosis factor-α [10,11]. The pathologic findings of AE-ILD patients showed a variety of pathological patterns such as organizing pneumonia or extensive fibroblast foci and diffuse alveolar damage in an acute stage [8]. AE-ILD showed a pathologically similar pattern to the ARDS. PMX-DHP, which was applied in ARDS and improved the patient’s prognosis, was considered for application to patients with AE-ILD for these reasons. It has been reported that the treatment with PMX-DHP can improve oxygen diffusion in ARDS patients [12,13]. Recently, PMX-DHP was reported to have beneficial effects on respiratory status and long-term outcomes in patients with an acute exacerbation of IPF (AE-IPF) [14,15,16,17,18]. Two comparative studies conducted in Japan compared the group treated with PMX-DHP and the group treated without PMX-DHP, and patients with AE-ILD who did not improve even after administration of high-dose steroids were enrolled. The results showed that the survival rate was better in the PMX-DHP group [19,20]. In one retrospective study, PMX-DHP improved the patient’s prognosis, and it showed that, the shorter the time from AE-IPF onset to PMX-DHP treatment, the better the patient’s prognosis [21].

Most of the previous studies were conducted in Japan and there is no clear evaluation of clinical outcomes. Although there is a study that recently reported the effect of PMX-DHP treatment in Korea, a small number of subjects were included [22]. So, this study was conducted to further investigate the change in oxygenation and serologic markers and clinical effects of PMX-DHP in AE-ILD patients in Korea.

## 2. Materials and Methods

### 2.1. Study Design and Patient Selection

We retrospectively examined the medical records of patients with AE of IPF or other types of ILD hospitalized at the tertiary academic hospital from January 2018 to October 2021. AE of IPF or ILD was defined according to the criteria suggested by Collard et al. [5] and other study criteria [15,19]: (1) acute worsening or development of respiratory symptom within 1 month; (2) new bilateral ground-glass opacities and/or consolidation on chest computed tomography; (3) PaO_2_/FiO_2_ ratio (P/F ratio) < 300 mmHg in the arterial blood gas analysis; and (4) absence of trauma, massive blood transfusion, pneumothorax, pulmonary thromboembolism, heart failure, and alternative causes of ARDS.

All the patient’s data were collected from the electronic medical records (EMR) (C&U care, Daejeon, Korea). Patient’s laboratory data and radiologic data were collected. Treatments administered during hospitalization and pre-hospitalization, length of hospital stay, and mortality data were collected from the EMR. In addition, the initial acute physiology and chronic health evaluation II (APACHE II) score was collected to evaluate the severity of the patient’s condition.

### 2.2. Treatment of PMX-DHP 

We administered PMX-DHP (PMX; Toray Medical Co., Ltd., Tokyo, Japan) to patients with AE-ILD receiving treatment with steroids alone or with cyclophosphamide. PMX-DHP treatment was considered when the patient’s P/F ratio was less than 300 or when the oxygen demand did not decrease or increase even after 24 h after the clinician performed standard treatment. A double-lumen catheter was inserted into the jugular or femoral vein. PMX-DHP was administered for 2 to 12 h (usually 6 h) at a flow rate of 100 mL/min and repeated once more within 24 h, if possible. Nafamostat mesilate was used as the anticoagulant.

### 2.3. Statistical Analysis

Continuous parameters are expressed as medians and interquartile ranges (IQRs). Wilcoxon tests are used to compare the changes in the laboratory data and vital sign between baseline and 24 h or 48 h after the first treatment of PMX-DHP. Univariate Cox regression analysis was performed to evaluate the association of 90-day mortality. The survival of patients with AE-ILD treated with PMX-DHP was analyzed using the Kaplan–Meier survival curve. *p*-values of <0.05 were considered statistically significant. SPSS software (version 22.0; IBM Corporation, Somers, NY, USA) was used to perform all statistical analysis.

## 3. Results

### 3.1. Patients’ Baseline Characteristics

During the study period, twenty-two patients with AE-ILD were treated with PMX-DHP. Table 1 showed the patient’s baseline characteristics. The median age of enrolled patients was 66 years (IQR: 60–77). Males made up 81.8% (18/22) of the patients, and the common underlying disease was hypertension 50.5% (11/22) and diabetes mellitus 36.4% (8/22). Of the enrolled patients, 50.0% (11/22) were diagnosed with IPF. The most used drug before the acute exacerbation was pirfenidone (27.3%, 6/22). 

The median APACHE II score was 17.5 (11.0–22.0). Sixteen patients had a previous pulmonary function test, with a median forced vital capacity (FVC) of 69.5% (% predicted, 54.0–79.3) and a median diffusing capacity of the lung for carbon monoxide (DLCO) of 53.5% (% predicted, 38.3–67.8).

### 3.2. Patient’s Treatment and Clinical Outcomes 

Table 2 showed patients’ treatment and outcomes. Methylprednisolone was used as a steroid pulse therapy. All patients were administered antibiotics with steroid therapy at the same time. PMX-DHP therapy was given within 48 h of AE of ILD for 72.7% (16/22) of patients. The 28-day mortality rate was 45.5% (10/22) and the 90-day mortality rate was 72.7% (16/22). The median duration of hospital stay was 21.0 (13.5–30.0) days from admission and 15.0 (7.3–26.0) days from the first PMX-DHP treatment (Figure 1). Invasive mechanical ventilation was applied in 59.1% (13/22) of cases, and the duration of ventilator application was 9.0 (5.0–16.5) days.

### 3.3. Vital Signs and Laboratory Data before and after PMX-DHP 

Table 3 showed changes in laboratory data and vital signs before and after the first PMX-DHP treatment. The P/F ratio significantly improved in patients from baseline to 24 h (median (IQR), 116.3 (88.5–134.3) mmHg vs. 168.6 (115.5–226.8) mmHg, *p* = 0.001), and 48 h (median (IQR), 116.3 (88.5–134.3) mmHg vs. 181.6 (108.9–232.0) mmHg, *p* = 0.003) (Figure 2A). The WBC count significantly decreased from baseline to 24 h (median (IQR), 13.9 (10.8–22.8) × 10^3^/µL vs. 10.3 (7.2–18.2) × 10^3^/µL, *p* = 0.001), and 48 h (median (IQR), 13.9 (10.8–22.8) × 10^3^/µL vs. 8.3 (5.1–16.4) × 10^3^/µL, *p* = 0.002) (Figure 2B). The CRP levels decreased from baseline to 48 h (median (IQR), 11.6 (3.8–20.7) mg/dL vs. 5.8 (2.9–10.7) mg/dL, *p* = 0.049) (Figure 2C). The interleukin-6 levels decreased from baseline to 24 h (median (IQR), 46.8 (8.8–277.8) pg/mL vs. 19.3 (6.2–51.9) pg/mL, *p* = 0.014) but did not statistically significantly decrease from baseline to 48 h (Figure 2D).

During PMX-DHP treatment, vital signs did not worsen (Table 3), and no patients required additional vasopressors. No additional bleeding was observed, and no transfusion was required during PMX-DHP treatment. There were no complications such as pneumothorax or hematoma related to catheterization. 

### 3.4. Factors Associated with Patients’ 90-Day Mortality

Univariate Cox regression analysis revealed factors associated with 90-day mortality (Table 4). Initial APACHE II scores were associated with 90-day mortality (odds ratio (OR), 1.267; 95% confidence interval (CI), 1.069–1.501; *p* = 0.006). PMX-DHP therapy within 48 h of AE of ILD was not associated with 90-day mortality.

## 4. Discussion

In this study, PMX-DHP treatment in patients with AE-ILD showed the potential to benefit, as in previous studies. After treatment with PMX-DHP, the P/F ratio improved and WBC, CRP, and IL-6 levels were decreased. The 28-day mortality rate was 45.5% and the 90-day mortality rate was 72.7% in the AE-ILD patients who underwent PMX-DHP treatment. There were no signs of deterioration of vital signs before and after PMX-DHP, and no severe complications occurred during PMX-DHP treatment.

In this study, oxygenation was improved after PMX-DHP treatment in AE-ILD patients. These results are similar to those of other previous studies. In the study of Seo et al., PMX-DHP was applied to IPF AE patients, and the alveolar–arterial oxygen pressure difference (A–a DO_2_) showed improvement in four out of six patients [14]. Oishi et al. compared the group with and without PMX-DHP in IPF AE patients, and the difference in the P/F ratio after treatment was improved in the PMX-DHP group compared with the non-PMX-DHP group (59.0 ± 15.9 vs. 2.2 ± 17.2, *p* = 0.044) [21]. Enomoto et al. applied the use of PMX-DHP in patients with AE-IPF. In the group in which PMX-DHP was used, the change in the P/F ratio was more improved than in the group in which PMX-DHP was not used (58.2 ± 22.5 vs. 0.7 ± 13.3, *p* = 0.034). In addition, the P/F ratio in the PMX-DHP group improved after 2 days of treatment compared to before treatment (*p* = 0.026) [20]. Hara et al. treated patients with rapid progressive ILD with PMX-DHP and showed an improvement in the P/F ratio and A–a DO_2_ after application [15]. In Korea, Lee et al. performed PMX-DHP treatment on ten AE-ILD patients, and it showed that the P/F ratio improved after treatment (86 (63–106) vs. 145 (86–260), *p* = 0.030) [22]. As shown in these studies, PMX-DHP treatment has the potential to help prognosis by improving oxygenation in AE-ILD patients.

In this study, some serological markers decreased after PMX-DHP treatment in AE-ILD patients. WBC, CRP, and IL-6 levels decreased after PMX-DHP. High WBC, CRP, and IL-6 levels in patients with acute exacerbation of interstitial lung disease are known to be associated with the patient’s prognosis [14,15,16,22,23,24,25,26,27]. When PMX-DHP treatment was performed in IPF AE patients, WBC was absorbed when PMX-DHP fibers were analyzed, and most of the cells were neutrophils [16]. MMP-9 was detected in the washed media of PMX. Additionally, blood matrix metalloproteinase-9 (MMP-9) levels were significantly decreased after the second PMX treatment compared to before PMX treatment. MMP-9 is known to contribute critically to lung tissue damage in IPF and ARDS. Adsorption by PMX of neutrophils producing activated MMP-9 or MMP-9 has therapeutic potential for IPF and ARDS. In Kamiya et al.’s study, high lactate dehydrogenase (LDH), WBC count and a high APACHE II score were associated with a poor prognosis for patients with IPF AE [27]. In a study by Yamazoe et al., high WBC count (OR 1.87; 95% CI 1.09–4.95; *p* = 0.01) and low hemoglobin (OR 0.26; 95% CI 0.04–0.78; *p* = 0.01) in patients with AE of IPF were associated with in-hospital mortality [23]. In a study by Hachisu et al. in patients with AE of IPF, high CRP (hazard ratio (HR) 1.080; 95% CI 1.022–1.141; *p* = 0.006), LDH (HR 1.003; 95% CI 1.000–1.006; *p* = 0.037) and low total cholesterol levels (HR 0.985; 95% CI 0.972–0.997; *p* = 0.018) were associated with in-hospital mortality [28]. In Papiris et al.’s study of patients with AE of IPF, IL-6 and IL-8 levels were high in acute exacerbation patients, and high IL-6 and IL-8 levels were associated with mortality [24]. In Lee et al.’s study of AE-ILD, high IL-6 was associated with an acute exacerbation and was an independent risk factor for mortality [25]. In Lee et al.’s study, the cut-off value for predicting the AE of IL-6 was 25.20 pg/mL (sensitivity 66.7%, specificity 80.6%). Based on the results from these studies [23,24,25,27,28], high WBC, CRP, and IL-6 were associated with an acute exacerbation of ILD and high mortality. A decrease in these serological markers after PMX-DHP treatment has the potential to have a positive effect on the patient’s prognosis. In other studies, MMP-9, Krebs von den Lungen-6, LDH, monocyte chemoattractant protein-1, growth-regulated peptide a, IL-6, and CRP decreased after PMX-DHP treatment in AE ILD patients [14,15,16,22]. In addition to the serological markers identified in this study, a reduction in other serological markers has the potential to help reduce fibrotic changes in the lungs of patients with AE-ILD and improve the prognosis.

In this study, the 28-day mortality rate was 45.5% and the 90-day mortality rate was 72.7% in patients with AE-ILD after PMX-DHP treatment. PMX-DHP treatment has shown survival benefits in patients with an acute exacerbation of IPF or rapid progressive ILD [19,20,21]. In patients treated with PMX-DHP for rapid progressive interstitial pneumonia, the 30-day mortality and 90-day mortality were 36.4% and 51.6%, respectively [15]. In AE-ILD patients treated with PMX-DHP, the 30-day and 90-day mortality rates were 27.3% and 72.7%, respectively [22]. The difference in mortality may have been influenced by the fact that the patients were not in the same interstitial lung disease group and did not have the same underlying condition. In this study, 90-day mortality was associated with the APACHE II score. In another study, PMX-DHP treatment in IPF-AE patients and changes in the P/F ratio and LDH before and after treatment were associated with survival [20,21]. Previous studies have shown that there is an association between mortality and the APACHE II score in interstitial lung disease patients admitted to the intensive care unit (ICU) [29,30,31].

This study has several limitations. First, as this is a single center study, the number of patients included in this study is small. This has been done with a small number of patients in previous studies as well. This is influenced by the small number of AE-ILD patients and, to overcome this, studies including a larger number of patients are likely to be needed in the future. Second, in this study, survival was not compared with the group of patients who did not receive PMX-DHP treatment. So, the prognosis could not be compared with the non-treatment patient group. Third, 13.6% (7/22) of patients were diagnosed with ILD by surgical biopsy. This is because, in many cases, the patient’s condition worsened, and surgical biopsy was often impossible to perform because of the patient’s condition. In cases of diagnosis only by radiology, the diagnosis was carried out under consultation with a radiologist and respiratory specialists. Fourth, PMX-DHP treatment was usually performed in critically ill patients who received a high-flow nasal cannula or invasive mechanical ventilation and were admitted to the ICU. So, it was impossible to confirm the change in oxygenation and serologic markers after PMX-DHP in AE-ILD patients who required relatively low oxygen.

## 5. Conclusions

In conclusion, PMX-DHP treatment improved the P/F ratio and decreased WBC and CRP levels in AE-ILD patients. No severe complications occurred during PMX-DHP treatment. Although survival benefit was not confirmed, improved oxygenation and changes in the serological marker findings suggest that it has the potential benefit of improving the prognosis for AE-ILD patients. Therefore, a large, randomized trial will be helpful to confirm the improvement in clinical course and the survival of AE-ILD patients after PMX-DHP treatment.

## Figures and Tables

**Figure 1 jcm-11-02485-f001:**
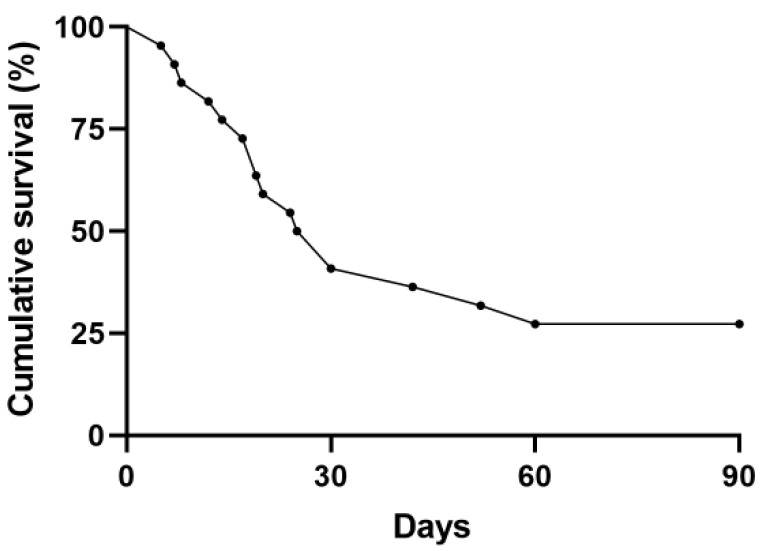
Kaplan–Meier survival curve for patients with acute exacerbation of interstitial lung disease treated with polymyxin B-immobilized fiber column (PMX-DHP). Among patients treated with PMX-DHP, the 28-day mortality rate was 45.5% and the 90-day mortality rate was 72.7%.

**Figure 2 jcm-11-02485-f002:**
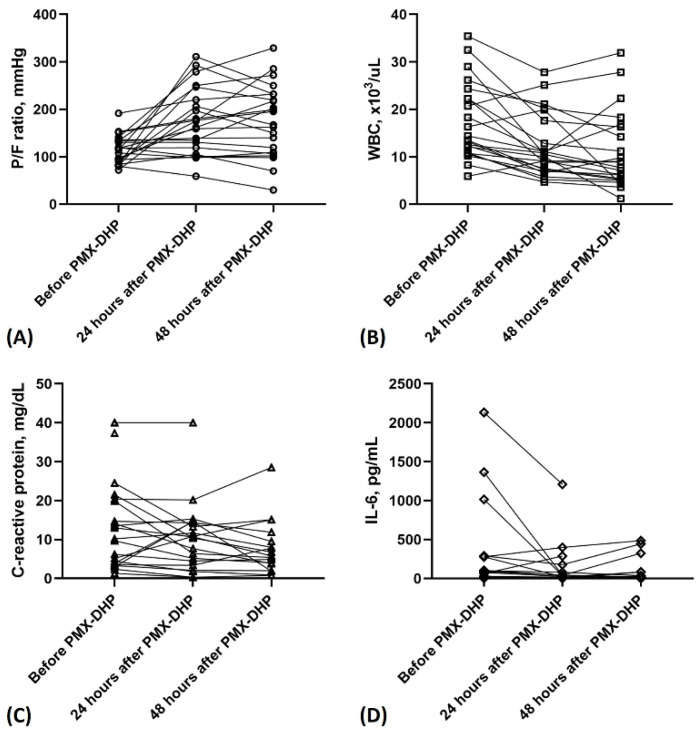
Changes in P/F ratio, white blood cell, c-reactive protein, and interleukin-6 levels before and after polymyxin B-immobilized fiber column (PMX-DHP) treatment. (**A**). P/F ratio, (**B**). White blood cell, (**C**). C-reactive protein, (**D**). Interleukin-6.

**Table 1 jcm-11-02485-t001:** Baseline characteristics of the patients.

Baseline Characteristics	Value
Age, years	66 (60–77)
Male (%)	18 (81.8)
Body mass index, kg/m^2^	23.8 (21.7–24.9)
Idiopathic pulmonary fibrosis	11 (50.0)
ILD excluding IPF	11 (50.0)
Underlying disease	
Hypertension	11 (50.0)
Diabetes mellitus	8 (36.4)
Solid tumor	3 (13.6)
Hematologic malignancy	1 (4.5)
Chronic kidney disease	1 (4.5)
Cerebrovascular accident	2 (9.1)
Medication prior to acute exacerbation	
Steroid	4 (18.2)
Pirfenidone	6 (27.3)
Cyclophosphamide	2 (9.1)
APACHE II score	17.5 (11.0–22.0)
Pulmonary function test (*n* = 16)	
FVC, % predicted	69.5 (54.0–79.3)
FEV1, % predicted	82.0 (61.8–93.3)
DLCO, % predicted	53.5 (38.3–67.8)

Data are presented as median and interquartile range or number (%), unless otherwise indicated. ILD: interstitial lung disease, IPF: idiopathic pulmonary fibrosis, APACHE II: acute physiology and chronic health evaluation II, FVC: forced vital capacity, FEV1: forced expiratory volume in one second, DLCO: The diffusing capacity for carbon monoxide.

**Table 2 jcm-11-02485-t002:** Treatment and outcome of the patients.

Treatment and Outcome	Value
Medical therapy used for AE	
Methylprednisolone	22 (100.0)
Antibiotics	22 (100.0)
PMX-DHP therapy within 48 h of AE of ILD	16 (72.7)
28-day mortality	10 (45.5)
90-day mortality	16 (72.7)
Invasive mechanical ventilation	13 (59.1)
Duration of ventilator, days	9.0 (5.0–16.5)
Hospital stay, days	21.0 (13.5–30.0)
Survival days from 1st PMX-DMP therapy	15.0 (7.3–26.0)

Data are presented as median and interquartile range or number (%), unless otherwise indicated. AE: acute exacerbation, PMX-DHP: direct hemoperfusion with polymyxin B immobilized fiber, ILD: interstitial lung disease.

**Table 3 jcm-11-02485-t003:** Clinical course of laboratory data before and after PMX-DHP treatment based on the Wilcoxon test.

Value	Baseline		24 h			48 h		
	Median, IQR	*n*	Median, IQR	*n*	*p*-Value	Median, IQR	*n*	*p*-Value
Lab								
pH	7.43 (7.35–7.47)	22	7.43 (7.37–7.46)	22	0.944	7.42 (7.37–7.45)	22	0.987
PaCO_2_, mmHg	41.0 (34.8–49.8)	22	39.5 (34.8–48.0)	22	0.134	39.5 (35.5–44.0)	22	0.158
PaO_2_, mmHg	74.0 (63.8–99.3)	22	102.5 (79.8–126.8)	22	0.046	110.0 (86.5–120.5)	22	0.011
P/F ratio, mmHg	116.3 (88.5–134.3)	22	168.6 (115.5–226.8)	22	0.001	181.6 (108.9–232.0)	22	0.003
WBC, ×10^3^/µL	13.9 (10.8–22.8)	22	10.3 (7.2–18.2)	22	0.001	8.3 (5.1–16.4)	22	0.002
Hb, g/dL	11.9 (10.5–13.6)	22	10.7 (9.8–13.0)	22	0.020	11.1 (9.5–12.7)	22	0.011
Platelet, ×10^3^/µL	239.0 (180.3–279.5)	22	191.5 (122.8–248.8)	22	<0.001	167.5 (74.0–234.0)	22	<0.001
CRP, mg/dL	11.6 (3.8–20.7)	22	10.6 (3.7–14.5)	20	0.159	5.8 (2.9–10.7)	17	0.049
IL-6, pg/mL	46.8 (8.8–277.8)	22	19.3 (6.2–51.9)	22	0.014	24.7 (2.3–144.8)	14	0.778
Vital sign								
Mean BP, mmHg	88.5 (82.8–95.0)	22	82.5 (78.0–92.5)	22	0.445	86.0 (78.8–93.5)	22	0.626
Heart rate, beats/min	99 (86–120)	22	94 (82–114)	22	0.051	91 (71–117)	22	0.041
Respiratory rate, beats/min	25 (22–28)	22	23 (20–27)	22	0.808	24 (20–29)	22	0.191
Body temperature, °C	37.1 (36.8–37.3)	22	36.9 (36.6–37.3)	22	0.296	36.8 (36.5–37.0)	22	0.012

Data are presented as median and interquartile range. IQR: interquartile range, pH: potential hydrogen, PaCO_2_: partial pressure or carbon dioxide, PaO_2_: partial pressure of oxygen, P/F ratio: ratio of arterial oxygen partial pressure to fractional inspired oxygen, WBC: white blood cell, Hb: hemoglobin, CRP: C-reactive protein, IL-6: interleukin-6, BP: blood pressure.

**Table 4 jcm-11-02485-t004:** Univariate Cox regression analysis results on 90-day mortality.

Parameters	Odds Ratio	95% Confidence Interval	*p*-Value
Age	1.056	0.968–1.151	0.218
Male	4.023	0.614–26.378	0.147
Body mass index, kg/m^2^	0.664	0.439–1.005	0.053
Initial APACHE II scores	1.267	1.069–1.501	0.006
Initial laboratory data			
White blood cell	0.982	0.882–1.095	0.747
C-reactive protein	0.982	0.922–1.046	0.576
Interleukin-6	0.999	0.998–1.001	0.397
P/F ratio	0.951	0.902–1.002	0.061
PMX-DHP therapy within 48 h of AE of ILD	1.212	0.200–7.336	0.834

APACHE II: Acute Physiology and Chronic Health Evaluation II, P/F ratio: ratio of arterial oxygen partial pressure to fractional inspired oxygen, PMX-DHP: direct hemoperfusion with polymyxin B immobilized fiber, AE: acute exacerbation, ILD: interstitial lung disease.

## Data Availability

Not applicable.

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
