# Peer review of "Changes in Oxygenation and Serological Markers in Acute Exacerbation of Interstitial Lung Disease Treated with Polymyxin B Hemoperfusion"

_jcm, 2022, doi:10.3390/jcm11092485_

Round 1
Reviewer 1 Report
Overall, this is a good non-randomized study on a very-very important topic which is AE-ILD.
Would recommend the following revisions:
1) While this topic is very important, the lack of a comparison arm is a major problem. For example, the improvement in the P to F ratio could be attributed to other treatments such as positive pressure ventilation, steroids, diuretics etc. Most of the endpoints such as the WBC count and CRP levels are not very influential, and they were not even significant in your regression model. The mortality overall seems not much different than other patients with an acute exacerbation of ILD. As such I would recommend a retrospective comparison to patients that have not been treated with PMX from the prior years at your hospital.
If this cannot be done, the article can be accepted but many things will need to be rephrased to focus on "biomarkers in AE-ILD and PMX-DHP", rather than efficacy.
If you cannot compare to prior non PMX controls (ok to not be from the same years in the hospital), then rephrase many parts of the title and discussion sections to focus on just "secondary markers" rather than true efficacy of PMX. Also spend some more time explaining why mortality did not seem much changed after PMX compared to the general literature.
2) In the abstract, change the last conclusion:
"In patients with AE-ILD, PMX-DHP treatment was associated with improved P/F ratio and lower WBC, CRP levels".
3) in the introduction I think it is important to use 1 or 2 sentences to further expand on the biochemical mechanism of PMX-DHP
Author Response
Reviewer: 1
Overall, this is a good non-randomized study on a very-very important topic which is AE-ILD.
Thank you for your comments. We did our best to answer all the questions and comments raised by reviewers.
Would recommend the following revisions:
1) While this topic is very important, the lack of a comparison arm is a major problem. For example, the improvement in the P to F ratio could be attributed to other treatments such as positive pressure ventilation, steroids, diuretics etc. Most of the endpoints such as the WBC count and CRP levels are not very influential, and they were not even significant in your regression model. The mortality overall seems not much different than other patients with an acute exacerbation of ILD. As such I would recommend a retrospective comparison to patients that have not been treated with PMX from the prior years at your hospital.
If this cannot be done, the article can be accepted but many things will need to be rephrased to focus on "biomarkers in AE-ILD and PMX-DHP", rather than efficacy.
If you cannot compare to prior non PMX controls (ok to not be from the same years in the hospital), then rephrase many parts of the title and discussion sections to focus on just "secondary markers" rather than true efficacy of PMX. Also spend some more time explaining why mortality did not seem much changed after PMX compared to the general literature.
Authors’ Response : Thank you for your comments. Since polymyxin B hemoperfusion was mainly performed in patients with acute exacerbation of ILD with high oxygen demand, we could not find a control group and make it a comparative group. As commented by reviewer, we revised the title and discussion section as below.
Revised manuscript (Title)
Changes of oxygenation and serological markers in acute exacerbation of interstitial lung disease treated with polymyxin B hemoperfusion
Revised manuscript (Discussion)
In this study, some serological markers decreased after PMX-DHP treatment in AE-ILD patients. WBC, CRP, and IL-6 levels decreased after PMX-DHP. High WBC, CRP, and IL-6 levels in patients with acute exacerbation of interstitial lung disease are known to be associated with the patient's prognosis [1-5]. [6-9]. When PMX-DHP was performed in IPF AE patients, WBC was absorbed when PMX-DHP fibers were analyzed, and most of the cells were neutrophils [8]. MMP-9 was detected in the washed media of PMX. Additionally, blood MMP-9 levels were significantly decreased after the second PMX treatment compared to before PMX treatment. Matrix metalloproteinase-9 (MMP-9) is known to contribute critically to lung tissue damage in IPF and ARDS. Adsorption by PMX of neutrophils producing activated MMP-9 or MMP-9 has therapeutic potential for IPF and ARDS. In Kamiya et al.'s study, high lactate dehydrogenase (LDH), WBC count and high APACHE II score were associated with prognosis in patients with IPF AE [5]. In a study by Yamazoe et al., high WBC count (OR 1.87; 95% CI 1.09 - 4.95; p = 0.01) and low hemoglobin (OR 0.26; 95% CI 0.04 - 0.78; p = 0.01) in patients with AE of IPF were associated with in-hospital mortality [1]. In a study by Hachisu et al. in patients with AE of IPF, high CRP (hazard ratio [HR] 1.080; 95% CI 1.022 - 1.141; p = 0.006) , LDH (HR 1.003; 95% CI 1.000 - 1.006; p = 0.037) and low total cholesterol levels (HR 0.985; 95% CI 0.972 - 0.997; p = 0.018) were associated with in-hospital mortality [10]. In Papiris et al.'s study of patients with AE of IPF, IL-6 and IL-8 levels were high in acute exacerbation patients, and high IL-6 and IL-8 levels were associated with mortality [2]. In Lee et al.'s study of AE-ILD, high IL-6 was associated with acute exacerbation and was an independent risk factor for mortality [3]. In Lee et al.’s study, the cut-off value for predicting the AE of IL-6 was 25.20 pg/mL (sensitivity 66.7%, specificity 80.6%). Based on the results from these studies [1-3, 5, 10], high WBC, CRP, and IL-6 were associated with acute exacerbation of ILD and high mortality. A decrease in these serological markers after PMX-DHP treatment has the potential to have a positive effect on the patient's prognosis. In other studies, matrix metalloproteinase-9, krebs von den lungen-6, LDH, monocyte chemoattractant protein-1, growth-regulated peptide a, IL-6, and CRP decreased after PMX-DHP in AE ILD patients [6-9]. In addition to the serological markers identified in this study, reduction in other serological markers has the potential to help reduce fibrotic changes in the lungs of patients with AE-ILD and improve the prognosis.
2) In the abstract, change the last conclusion:
"In patients with AE-ILD, PMX-DHP treatment was associated with improved P/F ratio and lower WBC, CRP levels".
Authors’ Response : We are sorry to make confusion. As commented by reviewer, we revised the abstract section as below.
Revised manuscript (Abstract)
Conclusions: In patients with AE-ILD, PMX-DHP treatment was associated with improved the P/F ratio and lower WBC, CRP levels.
3) in the introduction I think it is important to use 1 or 2 sentences to further expand on the biochemical mechanism of PMX-DHP
Authors’ Response : Thank you for your comments. As commented by reviewer, we revised the introduction section as below.
Revised manuscript (Introduction)
Polymyxin B direct hemoperfusion (PMX-DHP) is a medical device using polystyrene fibers and originally developed to remove endotoxin from endotoxemia observed in sepsis caused by Gram-negative bacilli [11], and in acute respiratory distress syndrome (ARDS). PMX-DHP is also effective in reducing several other serological markers such as interleukin (IL)-6, IL-9, IL-12, IL-17, platelet-derived growth factor, vascular endothelial growth factor and tumor necrosis factor-α [12, 13]. The pathologic findings of AE-ILD patients showed a variety of pathological patterns such as organizing pneumonia or extensive fibroblast foci and diffuse alveolar damage in an acute stage [14]. AE-ILD showed a pathologically similar pattern to the ARDS. PMX-DHP which was applied in ARDS and improved the patient's prognosis, was considered for application to patients with AE-ILD for these reasons.

Reviewer 2 Report
The present work entitled "Polymyxin B hemoperfusion improves oxygenation in acute exacerbations of interstitial lung disease" demonstrates some important points of reflection and above all it states that the use of Polymyxina B could be useful in improving the survival of patients with pulmonary interstitial disease.
However, several critical points emerge:
1) In the methodology of the work there are no control groups and the patients are too few to be able to relely assert the beneficial effects of the therapy.
2) Furthermore, in a good part of the patients the diagnosis was not obtained with biopsy examinations. This influences the actual underlying pathology and could be a confounding factor. Polymyxin for which pathology could it be really useful ?.
3) Claiming that IL-6 is reduced as well as CRP and WBC is too simplistic to define such improvements as a real long-term response to therapy.
In this regard, the paper could take on a more meaningful value if it were presented as a feasibility study or as a pilot study in anticipation of broader protocols and with control groups (perhaps randomized).
Another critical point is the lack of lung function values ​​(FEV1, FVC and Dlco) of these patients after treatment. Most of these patients are in fact monitored with respiratory function tests and the response to the therapies in place uses these parameters in the follow-up. Furthermore, even the basal values ​​are not present in all the patients under examination.
For these reasons I suggest a substantial revision, if possible, of what has been presented.
Also improve native English.
Author Response
The present work entitled "Polymyxin B hemoperfusion improves oxygenation in acute exacerbations of interstitial lung disease" demonstrates some important points of reflection and above all it states that the use of Polymyxin B could be useful in improving the survival of patients with pulmonary interstitial disease.
Thank you for your comments. We did our best to answer all the questions and comments raised by reviewers.
However, several critical points emerge:
1) In the methodology of the work there are no control groups and the patients are too few to be able to relely assert the beneficial effects of the therapy.
Authors’ Response : We are sorry to make confusion. We usually performed PMX-DHP treatment for patients who applied high flow nasal cannula or invasive mechanical ventilation for acute exacerbation of interstitial lung disease, so the number of patients was small. So, we changed the title to better express the paper. As commented by reviewer, we revised the title and limitation section as below.
Revised manuscript (Title)
Changes of oxygenation and serological markers in acute exacerbation of interstitial lung disease treated with polymyxin B hemoperfusion
Revised manuscript (Limitation)
In case of diagnosis only by radiology, the diagnosis was carried out under consultation with radiologist and respiratory specialists. Fourth, PMX-DHP treatment was usually performed in critically ill patients who received high flow nasal cannula or invasive mechanical ventilation admitted to an intensive care unit. So, it was impossible to con-firm the change of oxygenation and serologic marker after PMX-DHP in AE-ILD pa-tients who required relatively low oxygen.
2) Furthermore, in a good part of the patients the diagnosis was not obtained with biopsy examinations. This influences the actual underlying pathology and could be a confounding factor. Polymyxin for which pathology could it be really useful ?.
Authors’ Response : Thank you for your comments. The pathologic findings in patients with ILD AE may have a diverse pathological pattern with either an acute pattern of OP or extensive fibroblastic foci and DAD. It is pathologically similar to the tissue damage pattern in acute respiratory distress syndrome (ARDS). Therefore, PMX-DHP, which has been effective in ARDS, has started to be applied to patients with ILD AE. As commented by reviewer, we revised the introduction section as below.
Revised manuscript (Introduction)
Polymyxin B direct hemoperfusion (PMX-DHP) is a medical device using polystyrene fibers and originally developed to remove endotoxin from endotoxemia observed in sepsis caused by Gram-negative bacilli [11], and in acute respiratory distress syndrome (ARDS). PMX-DHP is also effective in reducing several other serological markers such as interleukin (IL)-6, IL-9, IL-12, IL-17, platelet-derived growth factor, vascular endothelial growth factor and tumor necrosis factor-α [12, 13]. The pathologic findings of AE-ILD patients showed a variety of pathological patterns such as organizing pneumonia or extensive fibroblast foci and diffuse alveolar damage in an acute stage [14]. AE-ILD showed a pathologically similar pattern to the ARDS. PMX-DHP which was applied in ARDS and improved the patient's prognosis, was considered for application to patients with AE-ILD for these reasons. It has been reported that the treatment with PMX-DHP can improve oxygen diffusion in ARDS patients [15, 16]. Recently, PMX-DHP was reported to have beneficial effects on respiratory status and long-term outcome in patients with acute exacerbation of IPF (AE-IPF) [6, 8, 9, 17, 18].
3) Claiming that IL-6 is reduced as well as CRP and WBC is too simplistic to define such improvements as a real long-term response to therapy.
Authors’ Response : The improvement of oxygenation gives the patient time to recover, so it can be said that he has an opportunity for recovery of the lungs. In addition, it is thought that the reduction of inflammatory markers can help prognosis by reducing damage to the lungs of patients.
Revised manuscript (Discussion)
In other studies, matrix metalloproteinase-9, krebs von den lungen-6, LDH, monocyte chemoattractant protein-1, growth-regulated peptide a, IL-6, and CRP decreased after PMX-DHP in AE ILD patients [6-9]. In addition to the serological markers identified in this study, reduction in other serological markers has the potential to help reduce fibrotic changes in the lungs of patients with AE-ILD and improve the prognosis.
In this regard, the paper could take on a more meaningful value if it were presented as a feasibility study or as a pilot study in anticipation of broader protocols and with control groups (perhaps randomized).
Another critical point is the lack of lung function values ​​(FEV1, FVC and Dlco) of these patients after treatment. Most of these patients are in fact monitored with respiratory function tests and the response to the therapies in place uses these parameters in the follow-up. Furthermore, even the basal values ​​are not present in all the patients under examination.
For these reasons I suggest a substantial revision, if possible, of what has been presented.
Also improve native English.
Authors’ Response : In our study, it is difficult to confirm the effect on prognosis improvement through control evaluation as there is no control group. However, in the absence of a specific treatment that can improve the patient's prognosis in acute exacerbation of interstitial lung disease, treatment methods that improve oxygenation and reduce inflammatory markers seem to be helpful for the patient's prognosis. It would have been better to be able to check the before and after changes in the lung function test, but there were many cases where lung function could not be performed due to sudden deterioration of the patient, and it was not possible to follow up the lung function test due to the death of the patient. As the reviewer mentioned, there were some lacking points in the title and content of this paper, so we tried to improve it. Accordingly, the title was changed to "Changes of oxygenation and biomarkers in acute exacerbation of interstitial lung disease treated with polymyxin B hemoperfusion", and the entire manuscript was revised, and make efforts to improve English. We underwent Editage (www.editage.co.kr) for English language editing.

Round 2
Reviewer 2 Report
The authors have shown a great commitment to revise the entire paper following the suggestions of the reviewers.
All comments are satisfactory.
It can be accepted for publication.